# Colorectal Cancer in Elderly Patients with Surgical Indication: State of the Art, Current Management, Role of Frailty and Benefits of a Geriatric Liaison

**DOI:** 10.3390/ijerph18116072

**Published:** 2021-06-04

**Authors:** Nicolás M. González-Senac, Jennifer Mayordomo-Cava, Angela Macías-Valle, Paula Aldama-Marín, Sara Majuelos González, María Luisa Cruz Arnés, Luis M. Jiménez-Gómez, María T. Vidán-Astiz, José Antonio Serra-Rexach

**Affiliations:** 1Geriatric Department, Hospital General Universitario Gregorio Marañón, 28007 Madrid, Spain; nic.gsenac@gmail.com (N.M.G.-S.); jennifer.mayordomo@gmail.com (J.M.-C.); paula.aldama.marin@gmail.com (P.A.-M.); saramajuelos@gmail.com (S.M.G.); mcarnes@salud.madrid.org (M.L.C.A.); joseantonio.serra@salud.madrid.org (J.A.S.-R.); 2Instituto de Investigación Sanitaria Gregorio Marañón, 28007 Madrid, Spain; 3Biomedical Research Networking Center on Frailty and Healthy Aging, CIBERFES, 28029 Madrid, Spain; 4School of Physical Activity and Sport Sciences, Universidad Politécnica de Madrid, 28040 Madrid, Spain; angela.macias.valle@alumnos.upm.es; 5Centro Asistencial San Camilo de Tres Cantos, 28760 Madrid, Spain; 6General Surgery Department, Hospital General Universitario Gregorio Marañón, 28007 Madrid, Spain; luismiguel.jimenez@salud.madrid.org; 7School of Medicine, Universidad Complutense, 28040 Madrid, Spain

**Keywords:** colorectal cancer, elderly, frailty, geriatric syndromes, comprehensive geriatric assessment, geriatric liaison, multicomponent programs, functional capacity

## Abstract

Six out of every 10 new colorectal cancer (CRC) diagnoses are in people over 65 years of age. Current standardized surgical approaches have proved to be tolerable on the elderly population, although post-operative complications are more frequent than in the younger CRC population. Frailty is common in elderly CRC patients with surgical indication, and it appears to be also associated with an increase of post-operative complications. Fast-track pathways have been developed to assure and adequate post-operative recovery, but comprehensive geriatric assessments (CGA) are still rare among the preoperative evaluation of elderly CRC patients. This review provides a thorough study of the effects that a CGA assessment and a geriatric intervention have in the prognosis of CRC elderly patients with surgical indication.

## Highlights

(1)Individualizing interventions in the colorectal cancer elderly population undergoing elective surgery could boost the benefits of fast-track pathways (as ERAS program), which may be gained by CGA-guided care and a geriatric liaison or co-management.(2)CGA should be the first step towards a multidisciplinary network which would give the patient access to a personalized pre-habilitation and rehabilitation program.

## 1. Background

### 1.1. Epidemiology

Colorectal cancer (CRC) is the third most common cancer in the world. In 2019, the worldwide number of new CRC cases was 1,931,590, which accounts for 10% of new cancer diagnosis. The median age of CRC diagnosis is 67 years, with 56% of the cases newly diagnosed corresponding to patients ≥65 years, and 31% to patients ≥75 years [1]. The median age at death is 72 years, and 45% of the deaths occur in patients ≥75 years old, with 21% of them in the oldest (≥85 years) [1]. The incidence of CRC has increased in countries with a medium-high human development index (HDI), whilst it has stabilized—or even declined—in some of the highest HDI, such as the United States [2] or certain European countries, possibly linked to the effect of screening CRC programs [3], changes in lifestyle and dietary habits [4,5]. Although more than 90% of colorectal carcinomas are adenocarcinomas, other rare types include neuroendocrine, squamous cell, adenosquamous, spindle cell and undifferentiated carcinomas [6].

### 1.2. Risk Factors

Main risk factors for the development of CRC are positive family history [7], male sex and advanced age, although lifestyle-related factors such as smoking, processed/red meat and alcohol intake, low-fruit and vegetable diets and increased bodyweight are also of importance [5,8].

### 1.3. Screening for Colorectal Cancer

The lack of specificity in CRC symptoms makes screening highly relevant, and many tests have been developed over the years (e.g., stool test, colonoscopy). It is generally recommended to give patients the opportunity to choose the test of their preference, as that may increase adherence to the screening program [9]. It appears that the early detection by screening may have contributed to the reduction of mortality by decreasing incidence (removing precancerous polyps) and increasing survival (detecting the disease at an early stage) [8,10]. Since the early 2000s a decrease in the elective and emergency admission rates for CRC resection (8% and 6%, respectively) has been observed in the U.S. aging population (≥65 years) [11].

Screening is carried out mainly in patients between 60 and 70 years of age [12], although it is recommended for subjects of 50 years and older [13]. A large proportion of patients 80 years and older still require urgent admissions, which could be related to the possible lack of screening in elderly patients. Moreno et al. described how only 4% of CRC diagnosis in elderly patients (>75 years) of a medical institution in the U.S. were determined through screening colonoscopy, whereas in younger patients (range: 50–75 years) this percentage rose to 14% [14]. Both the U.S. Preventive Services Task Force and the American Cancer Society recommend CRC screening until the age of 75 years (when life expectancy is greater than 10 years) and individualization in patients between 76 to 85 years, as in this age group screening benefits decrease while the risk of suffering associated complications increases [9,15]. The U.S. Multi-Society Task Force of Colorectal Cancer suggests continuing screening up to 85 years only if no previous screening has been done, and stopping it at 75 years if prior screening tests have been negative [16]. Interestingly, Van Hees et al. carried out a microsimulation modeling study to try to determine at what age CRC screening should still be considered and they concluded that in unscreened elderly without comorbidity colonoscopy screening was cost-effective up to age 85 years, decreasing to 82 and 79 years in the case of elderly patients with moderate to severe comorbidities, respectively [17].

### 1.4. Colorectal Cancer Management

Endoscopic management is achievable in early malignant lesions, but surgery remains the main foundation of CRC treatment. According to the U.S. Healthcare Cost and Utilization Project Nationwide Inpatient Sample, comprised by 1,043,108 patients over 45 years of age that had undergone CRC resection, most of them (64%) were 65 years or older, including 29% septuagenarians and 23% octogenarians and nonagenarians [11]. Even though a mean decrease in mortality after CRC surgery was described in the early 2000s (being this improvement most notable in patients 85 years and older) [11], and that long-term survival is achieved in the surgical CRC elderly population [10], it has been stated in the literature that differences of CRC management exist between age groups. Elderly patients are less likely to undergo CRC surgery in comparison to younger individuals [18]. Simmonds et al. showed how, in a population of 34,194 patients with CRC from various studies, 21% of those aged over 85 years did not undergo operation, while the rates of no surgical intervention were 11% in the 75–84 years age group, 6% in the 65–74 group and 4% in those aged 64 years or younger. Additionally, 33% of the surgeries performed in patients over 85 years had a palliative intent [19].

Also, elderly patients with a more advanced tumor stage were less often offered adjuvant therapy [18]. These differences have been sustained by other groups. Sell et al. recently described, in a retrospective analysis of patients with colon adenocarcinoma who underwent surgical resection in an American hospital, how younger patients (aged 79 years or less) were more likely to receive adjuvant chemotherapy when compared to octogenarians, with rates of 48% vs. 9%, respectively. Although they found phenotypic tumor differences between groups (octogenarians presented with larger tumors but less extra-colonic spread) the difference in adjuvant therapy remained when analyzing patients of all stages and when excluding those with American Joint Committee on Cancer Stage IV disease. Interestingly, they also described how in younger patients the use of chemotherapy increased with tumor size, while in the elderly it decreased [20]. Serra–Rexach et al. conducted a retrospective cohort study in a Spanish university hospital in which it was found that age was the main reason for different therapeutic approaches in elderly (≥75 years) and younger (<75 years) CRC patients. Although no differences were observed between groups in tumor degree of differentiation, extension or stage at diagnosis, those individuals aged over 75 years were less likely to receive surgery, radiotherapy, and chemotherapy [21]. Older population is frequently underrepresented in randomized clinical trials (RCTs) [22] and, consequently, it is difficult to reach evidence-based clinical recommendations that apply to the treatment of the elderly CRC population [23].

Laparoscopic surgery (LS) has gained importance over the years due to its short-term beneficial results compared to open colectomy (e.g., decreased post-operative morbidity, faster recovery of bowel function, reduction on length hospital of stay (LOS)) with a low rate of conversion to open surgery (OS) [24,25,26]. It has proven to be an effective and safe procedure for treating elderly CRC patients [27]. A matched case-control study by Hinoi et al. which only included elderly patients with a diagnosis of colon or rectum adenocarcinoma (median age: 83 years) pointed out how the outcomes of LS in this population are not inferior to those of OS [28]. Regarding the surgical act, they showed how the LS approach—for both colon and rectal cancer—was longer in duration, but had lower blood loss. In the post-operative period, patients who had undergone LS for colon cancer had a faster return of bowel function, a shorter hospital stay and were able to initiate fluid and solid diet in less time than OS patients. Those who had undergone rectum LS also had a faster return of bowel function and initiated fluid diet in less time than OS patients, but no differences were found regarding time to solid intake and hospital stay. Post-operative morbidity rates in colon cancer cases were 36% in the OS group and 25% in the LS group, at the expense of a reduction in the occurrence of delirium, organ/space surgical site infection and pneumonia. Although the post-operative morbidity rates in rectum patients were also lower (40% vs. 47%), the difference was not statistically significant [28]. Kannan et al. also observed how post-operative complications on elderly patients who had undergone laparoscopic partial colectomy were significantly fewer than in those that had undergone open partial colectomy, including every subcategory (cardiac, pulmonary, renal, and infectious). Additionally, the LS group had lower rates of unplanned return to the operating room, their LOS was shorter and 30-day mortality was also significantly lower [29].

Adjuvant therapy can be considered in high-risk Stage II patients (e.g., poorly differentiated tumor, vascular or perineural invasion, lymph nodes sampling <12, tumor presentation with obstruction or perforation) and is recommended in Stage III patients, as it improves survival [5,10]. As rectal cancer surgery is more complex, its approach tends to be different, and neoadjuvant radiotherapy is common [5].

## 2. Characteristics of Elderly Colorectal Cancer Population That Undergo Surgery

### 2.1. General Colorectal Cancer Characteristics

#### 2.1.1. Disease Presentation

Patients over 75 years old are more likely to present with later-stage disease and to undergo emergency surgery [19]. Bircan et al. retrospectively analyzed the characteristics of 265 patients that had undergone a programmed colorectal surgery at two Turkish institutions and found that the most common causes of admission differed between age groups: blood stool for patients aged 60–69 years, bowel obstruction for those aged between 70–79 years and anemia for the oldest (>80 years) [30]. Although these last findings could be due to the inclusion of non-malignant colon surgeries in the study, differences like these can also be attributed as stated by the Colorectal Cancer Collaborative Group [19] to age-related variances in recognizing symptoms or seeking medical advice, as well as to primary-care referral patterns.

#### 2.1.2. Complications and Post-Surgical Survival

Although intraoperative complications do not seem to be more frequent among elderly CRC patients [30,31,32], no clear consensus exists regarding surgical post-operative complications. Some authors have described that elderly patients suffer more ileus, peritonitis/septic shock, pelvic abscess, incisional/post-herniation and have significant longer time to first flatus, bowel motion or to resume normal diet compared to younger patients [31]. Others have not reported differences regarding post-operative surgical complications between age groups [30]. These last findings were also found in a prospective multicenter study conducted by the Colon/Rectum Cancer Working Group that included 19,080 surgical CRC patients: the rate of surgical post-operative complications was not higher in the elderly group (≥80 years) compared to the younger one [32].

Regarding systemic post-operative complications, all studies agree on how they tend to be more frequent in the elderly when compared to younger CRC patients. This type of complications includes respiratory [11,31,32], cardiovascular [11,31,32], renal [31,32], and infectious [11,31,32], among others. There is also certain consensus on how elderly CRC patients have longer LOS which may be attributed to, precisely, the higher rates of post-operative complications [11,30,31,32,33]. Kunitake et al. described an increase of 90-day post-discharge readmission rates in the elderly of a large CRC population (83,897 with colon cancer, 26,794 with rectal cancer). These readmissions were mainly justified by non-surgical complications and associated with higher comorbidity and male gender [33].

Although mortality after CRC surgery in the elderly has decreased in the past few years [11], age is an independent predictor of post-operative mortality following CRC resection [34,35]. Excess mortality is sustained throughout the whole year after CRC surgery, and most patients seem to die after the 30-day post-operative period, especially those aged 75 years or more [36]. Interestingly, older CRC patients who survive the first year after surgery may have the same overall cancer-related survival as younger patients [37].

### 2.2. Geriatric Syndromes

Geriatric syndromes (GS), such as cognitive impairment, functional dependency, falls or urinary incontinence, are clinical conditions more commonly detected on elderly patients. Its cause is believed to be multifactorial, and their presentation is the result of the accumulation of impairments in different systems and the inability of the individual to compensate for them. In the elderly population, GS are associated with higher risk of hospitalization and mortality. Both cancer and oncologic treatments can behave as potential stressors that may overwhelm the patient’s reserve capacity and, consequently, favor the development of GS. Therefore, the assessment of GS in the elderly cancer population is of interest when designing care plans or interventions [38].

#### 2.2.1. Functional Dependency

Functional dependency, understood as a person’s inability to live independently and perform basic activities of daily living, has proved to be a predictor of morbidity and mortality in the elderly population [39]. It has also been independently associated with shorter survival time in cancer patients [40].

In a systematic review by Hamaker et al. that gathered 23 studies which assessed long-term physical and role functioning changes in CRC patients after treatment, it was discovered that both physical and role functioning were significantly limited at three months after treatment [41]. Ronning et al. in an observational prospective cohort that evaluated predictors of postoperative complications in older patients with CRC, found a significant decline in both basic activities of daily living (ADLs) and instrumental activities of daily living (IADLs) in the 16–28 months that followed surgery, measuring ADLs and IADLs with Barthel Index and the Nottingham Extended Activities of Daily Living Scale, respectively [42].

#### 2.2.2. Frailty

Frailty is a dynamic clinical state characterized by an increased vulnerability to stressors that leads to a loss of homeostasis and a subsequent increase in the risk of developing adverse outcomes (such as disability, falls, delirium or death) [43,44]. Traditionally, it has been defined by two different models: the phenotype model [45] and the deficit accumulation model [46], both of which have showed overlap in their identification of frailty and statistical convergence [43]. The prevalence of frailty among community-dwelling older adults is variable, between 11% in subjects over 65 years of age (ascending to 16% in those aged 80–84 years and 26% in those aged ≥85 years) [47]. Regarding cancer patients, in a sample of 2349 Medicare beneficiaries with 65 years or more and a history of cancer the prevalence of frailty ranged from 46% to 80% [48]. It has been described in the literature how cancer patients and those who are undergoing surgery are more likely to be frail and have more adverse outcomes than those who are not frail [49]. As a result of this, oncologic scientific societies like the International Society of Geriatric Oncology (SIOG) recommend the screening of frailty in older cancer patients [50]. However, there is no standard evaluation and several tools to identify frail cancer patients have been developed, such as the Balducci criteria [51], the Vulnerable Elders Survey-13 [52] and the G8 Geriatric Screening Tool [53], among others. Many of these tools consider comorbidities, cognition, nutritional status, functionality, and physical performance as components of frailty, and others like the Fried criteria [45] only focus on a physical phenotype. Consequently, the prevalence of frailty in older individuals with CRC and surgical indication ranges between 25% and 46%, a variability that depends on both the population studied and the tools used to measure it [54]. It has been proposed that, regarding CRC treatment in the elderly, standard approaches may be offered to robust CRC patients while a need for an individualized therapeutic plan must be considered on frail CRC patients [55].

Using frailty as a risk-stratification tool in surgical elderly patients is a relatively new concept that could change their pre-operative assessment paradigm, as a growing body of scientific evidence has emerged over the past few years. Robinson et al., in a prospective cohort study that included patients ≥65 years undergoing elective colorectal or cardiac surgeries, described how those classified as frail had a higher risk of developing post-operative complications. The definition of frailty they used was based in a deficit accumulation model in which frailty was detected when at least 4 of 7 frailty-related characteristics (regarding function, cognition, chronic disease burden, walking speed, nutrition and geriatric syndromes) were met [56]. Independently of the way frailty is assessed, studies in CRC are concordant in the association of frailty and an increased risk of postoperative complications and mortality. Table 1 shows a detailed description of the main studies that have assessed the influence of frailty (defined by different criteria, such as the Groningen Frailty Indicator (GFI), the Fried Criteria and a modified version of the Balducci criteria) on post-operative outcomes in CRC patients, with similar results.

#### 2.2.3. Cognitive Impairment and Mental Health

The prevalence of dementia in patients with CRC is not clear. Gupta et al. in a population-level cohort study that included 17,507 patients of 67 years or more of the SEER-Medicare file diagnosed with colon cancer, found that the prevalence of dementia in newly-diagnosed patients was 7%. Also, they described how dementia patients were not only twice as likely to be diagnosed with non-invasive methods (without biopsy) but also twice as likely to have their cancer diagnosed after death [57].

The most frequently described psychological alterations in oncologic patients are reactive conditions, mainly adjustment disorder, followed by depressed mood and anxiety [58]. It has been described how depression prevalence in cancer patients can range from 0% to 58%, with this variability attributed to factors such as tumor type, stage of the disease, assessment instruments or diagnostic criteria employed [59]. Few data regarding mental illness in CRC patients have been published, but both depression and anxiety are common in this population, with published prevalence rates that range between 2–57% and 1–47%, respectively [60]. In a recent cohort study, Lloyd et al. described an increase in any mental illness (including depression, anxiety, and adjustment disorders, among others) in CRC survivors since diagnosis. They also found that risk factors for mental illness among CRC survivors include colostomy, female gender (for depression), radiation therapy, chemotherapy, older age, advanced disease, and comorbid conditions. CRC survivors who developed mental illness had increased mortality [61]. Regarding age, some studies have observed higher rates of depression in elderly CRC patients, but no difference on anxiety levels [62,63].

#### 2.2.4. Malnutrition and Social Support

Malnutrition prevalence in cancer population ranges from 20–70%, with differences attributed to patient’s age, cancer type and stage [64,65]. In the elderly cancer subpopulation, malnutrition has been described as a risk factor for mortality, functional decline and, among others, poor treatment response [66]. In a prospective multi-center study whose primary objective was to evaluate the impact of a geriatric screening and assessment in elderly patients with cancer (n = 1967, median age 76 years, 22% with CRC) it was found that, according to the Mini Nutritional Assessment, 68% of the patients were at risk of malnutrition and 15% had malnutrition. It was also found how, following the assessment, the most frequently planned geriatric intervention was related to nutrition (57%) [67].

Haviland et al. in a multicenter prospective cohort study that included 857 adult patients with CRC, found how—in a 2-year follow-up after surgery—levels of social support decreased over time and how health-related quality of life outcomes were associated with levels of social support. Also, their findings suggest that those patients with lower and declining social support were more likely to be older [68].

## 3. Multidisciplinary Team and Comprehensive Interventions in Colorectal Cancer Patients

To assure an adequate recovery after major abdominal surgery, fast-track pathways such as the Enhanced Recovery After Surgery (ERAS) program have been developed. Among other elements, ERAS programs include preoperative counselling, preferred laparoscopic approach, avoidance of nasogastric tubes or drains when not necessary, enforcement of postoperative early mobilization or feeding and detailed postoperative nursing-care programs [73,74]. It has been described how the implementation of at least four ERAS elements in the colorectal surgery pathway reduces LOS and the rate of post-operative complications without increasing readmission or mortality risk [75]. Its feasibility and benefits in elderly patients who undergo CRC surgery have already been described as well [76]. However, fast-track surgery programs do not discriminate between frail or robust elderly patients, which is of importance considering the higher rates of complications in the former collective. 

### 3.1. Benefits of a Geriatrics Liaison

The Comprehensive Geriatric Assessment (CGA) has been defined as a “multidimensional interdisciplinary diagnostic process intended to determine a frail elderly’s person’s medical, psychosocial and functional capabilities and limitations in order to develop an overall plan for treatment and long-term follow-up” [77]. It encompasses many domains of an elderly’s life to ensure the detection of a wide variety of problems (such as cognitive disorders, depression, social isolation, frailty, comorbidities, undernutrition, polypharmacy, and other geriatric syndromes) so that they can be properly managed to ensure the patient’s well-being and independence. 

To address the heterogeneity of elderly patients with cancer and guide oncologic treatment decisions, scientific societies such as the SIOG have recommended improving scientific research regarding CGA and cancer patients [78]. Although it has been described how CGA can contribute to the detection of problems and risks often unrecognized in regular oncologic assessments, or how CGA components have predictive risk of complications and toxicity related to treatment, a lack of standardized assessment tools and the heterogeneity regarding CGA models in geriatric oncology difficult its implementation [78]. Nevertheless, in the most recent American Society of Clinical Oncology Guideline for Geriatric Oncology, a CGA is recommended to all patients 65 years of age or older that are receiving chemotherapy [48], and many randomized controlled trials have shown that CGA-guided interventions improve key outcomes for older patients with cancer [79]. Unfortunately, its use in daily practice is complex and time-consuming. 

There are studies which show how a geriatric co-management (including both preoperative and postoperative care) in older patients undergoing cancer-related surgery is associated with a reduction of LOS [80] and a lower 90-day postoperative mortality [81]. As a result of this, individualizing interventions in the CRC elderly population undergoing elective surgery could boost the benefits of ERAS programs, which may be gained by CGA-guided care and a geriatric liaison or co-management. Interestingly, few surgeons appear to collaborate on a regular basis with geriatricians [82].

CGA-guided assessments in CRC patients can function as a risk assessment or as a tool to design individualized patient-centered interventions. Many studies have been described with conflicting results. Lee et al. carried out a retrospective review of a prospective single-center database to assess whether a preoperative CGA in 240 elderly patients (aged 70 years or more) who had undergone elective CRC surgery was effective in predicting postoperative morbidity. This CGA included several domains (comorbidity, polypharmacy, physical function, cognitive, depression and nutrition), and a “high-risk” patient was defined as one who had deficits in at least two of those domains. A total of 95 high-risk patients (40%) were detected, and this condition was significantly and independently associated with postoperative complications. As well, they found how greater independence in ADLs and fewer comorbidities were predictive of a less eventful—and therefore better—recovery [83].

Shipway et al. evaluated the efficacy of an embedded geriatric liaison service for emergency and elective gastrointestinal surgery using a retrospective control and a preoperative, in-hospital and post-operative CGA and intervention. The primary aim of the study was the reduction in LOS. They included 682 patients (203 pre-intervention, 479 post-intervention). A total of 132 patients in the intervention group were referred to the preoperative CGA-based assessment (from which 60% had CRC) and 26% of them were considered unfit and did not proceed with surgery. Two hundred and thirty-three inpatient reviews were conducted, being some of the most frequent indications discharge planning, communication with family, high dependency unit interventions, fluid balance, cardiac assessment, and delirium. The implementation of this geriatric liaison service supposed a mean LOS reduction of 3 days considering all surgeries, and this reduction was maintained in patients aged 75 years or more. However, when considering patients admitted electively for cancer surgery, LOS reductions were not statistically significant, although a trend of greater reduction was observed with advancing age [84]. Ramirez et al. carried out a before–after study with the objective of assessing the effect that a geriatric co-management program had on the LOS of elderly patients admitted to a general surgery ward. The results of the study show how in both intervention subgroups (the emergency-admitted and the electively-admitted), LOS was lower when compared to the control, with the CRC group presenting a mean decrease of 9 days [85].

In-hospital geriatric co-management interventions have also been described. In a single-center retrospective cohort study that included 310 patients aged 70 years or older who were admitted for elective CRC surgery in a tertiary level hospital, it was found how a daily CGA-based hospital assistance was associated with a lower incidence of delirium and other geriatric syndromes (such as falls, pain, urinary incontinence, constipation, pressure ulcers, malnutrition, and immobility), as well as fewer blood transfusions. Nevertheless, they also found that the intervention group had higher rates of long hospitalizations, intensive care unit admissions, serious complications, and hospitalization within the year [86].

However, although these kinds of CGA interventions seem promising, there are studies that have not shown clear benefits. Indrakusuma et al. carried out a retrospective matched-controlled study with the main objective of assessing beneficial postoperative outcomes of a preoperative CGA in 443 elderly patients (aged 70 years or more) who underwent CRC surgery in two time periods. The most frequent preoperative interventions derived from the preoperative CGA assessment were detection of delirium risk (64%), vitamin supplementation (64%) and dietary supplementation (20%). Contrary to benefits showed in the previously mentioned studies, in this study no differences regarding mortality, postoperative delirium or LOS were found when the intervention group was compared to the control group [87]. Similarly, an RCT of frail older patients that were to undergo CRC surgery, failed to detect benefits of a preoperative and tailored geriatric intervention focused on nutritional advice (34%), increased medication (30%), other healthcare professional referral (30%) and exercise (23%). No differences between groups were found regarding the rate of postoperative severe complications, reoperations, readmission, or mortality. However, they found that the intervention group experienced fewer medical non severe complications [70]. A detailed description of all these findings has been summarized in Table 2.

### 3.2. Benefits of Exercise Programs

Observational studies have shown how the highest level of physical activity (PA), before and after CRC diagnosis, is associated with a lower risk of CRC mortality [88,89,90]. Therefore, CRC patients should avoid physical inactivity whenever possible [91]. Although PA is beneficial in several aspects [92,93] its benefits vary depending on factors such as the population included, the study design or the exercise programs themselves [94].

We found eleven RCTs assessing the effect of physical exercise in surgical CRC patients over the age of 60 years (mean age range: 60–81 years), with sample sizes from 42 to 185 patients (Table 3). The RCTs included multimodal programs with prehabilitation (before surgery) [95,96,97,98]. prehabilitation and rehabilitation (after surgery) [99,100,101] or comparing prehabilitation vs rehabilitation [102,103,104,105]. Prehabilitation’s duration ranged between 2 and 4 weeks and rehabilitation from 4 to 8 weeks. Sessions were usually held one to three days per week for 30–60 min. Programs included exercise training alone [95,97,103], exercise training and psychological interventions with anxiety reduction strategies [98,99,102,103,104,105], adding in some of them a nutritional control [95,98,99,101,102,103,104,105]. Exercises included aerobic training [98], aerobic and resistance training [95,97,99,100,101,102,103,104,105], or only resistance training [96]. Aerobic training consisted of activities such as walking, treadmill walking or cycling, following the 150 min/week recommendations by the American College of Sport Medicine Guideline [88,91]. Resistance training consisted of exercises of the upper and lower limbs using elastic bands. Some of the RCTs did not report the exercise’s intensity, but most of them opted for a moderate one [88,91] using the Borg Scale [97,99,102,103,104]. Home-based exercise sessions supervised weekly with phone calls were more prevalent [96,98,99,102,103,104,105], although some programs were carried out at the hospital or exercise center [97,100,101]. As shown in Table 3, the results were variable. Some of the studies demonstrate an increase in functional capacity [99,104,105], and others a decrease in postoperative complications [95,101] or LOS [100,101]. Only two RCTs studied mortality, but no decrease was found in 30-day [100] or 1-year mortality [101] with the intervention. No differences regarding quality of life were found [97,104].

It is difficult to determine which is the most appropriate type of exercise program for elderly CRC population due to its heterogeneity [106]. The design of these programs must be adapted to the characteristics of the population studied (regarding functional capacity, frailty, comorbidity, etc.), the proposed outcomes (e.g., functional capacity recovery, change in functional state, reduction on complications or mortality, shortened LOS), the type of program (timing, duration, intensity…) and patient´s preferences.

### 3.3. Benefits of Psychotherapeutic Interventions

Psychotherapeutic interventions have been described in CRC population, mainly in patients with newly-formed stomas, with generally satisfactory effects. Stoma patients, due to a distorted body image and the loss of an essential body function, face difficulties in everyday life in terms of physical, psychological, and social aspects [107]. One of the most common psychosocial intervention described in this population is preoperative education, which appears to satisfactorily reduce both LOS and days to stoma proficiency [108,109]. Moreover, cognitive therapy and emotional support interventions such a relaxation training has proven to be feasible (even with follow-up telephone calls) and appear to reduce anxiety levels [110]. However, a major limitation is the small sample sizes of the studies [111].

Beneficial effects on quality of life with psychosocial interventions (mainly face-to-face approaches) [112] have been described [113]. Also, interventions focused on enriching communication with CRC patients have also been carried out. Ohlen et al. designed a person-centered information and communication intervention to study its beneficial effects on the patients’ preparedness for surgery, discharge, and their subsequent recovery, but no conclusive results were obtained [114].

A favorable trend on the effectiveness of a positive emotion-based psychological therapy and a cognitive-behavioral therapy on the quality of life of CRC patients receiving adjuvant chemotherapy has been suggested, although findings were not conclusive [115].

Ellis et al. in a report of a longitudinal study of psychological adjustment in 326 patients with advanced cancer (43% corresponding to CRC), found how elderly patients (aged 70 years or more) were less frequently referred for specialized psychosocial care in comparison to younger subjects with the same degree of depressive symptoms [116]. Whether this finding is related to the illness being less destabilizing to an older person when compared to a younger counterpart or to a possible ageist bias for which caregivers may assume the benefits of these therapies is greater in younger subjects, remains unknown.

The shortage of evidence regarding the effect of psychotherapeutic interventions in cancer patients is striking considering how receiving a cancer diagnosis is a complex and stressful experience that constitutes a vital crisis that can present itself with very diverse emotional reactions (e.g., sadness, anger, confusion) [117]. Future studies should focus on studying some of the mostly inconclusive findings mentioned above, adhere to standards of quality research, and try to, not only increase the number of individuals per study, but also focus on certain subpopulations, such as the elderly.

## 4. Conclusions

CRC is a frequent disease among the elderly. Although fast-track circuits such as the ERAS program include the assessment of relevant problems for the elderly CRC patient (e.g., malnutrition), other aspects that could influence therapeutic approaches are normally left out, such as functionality, frailty, cognitive impairment, depression/anxiety or social support. The adequate evaluation of these conditions could lead to its control or improvement, and therefore a change in the patient’s prognosis.

CGA has proven to be a useful tool for the identification and assessment of these conditions. It allows the multidisciplinary team (conformed by surgeons, anesthesiologists, nutritionists, pharmacologists, physical therapists, nurses, etc.) to design a thorough care plan that comprises both the oncologic treatment (surgery and/or adjuvant therapy) and the approach of geriatric syndromes through a multicomponent program. A program of these characteristics would be individualized, adjusted to the patient’s situation and preferences, and could include nutritional, psychotherapeutic, pharmacologic or exercise interventions, among others. Possible outcomes to be assessed would include not only length of stay, in-hospital mortality, or post-operative complications, but also improvement on the physical, functional, cognitive and mental health situation, quality of life and readmissions in both the medium and long term post-operative period.

The scarcity of randomized controlled trials that evaluate the benefits of preoperative geriatric assessments or the use of multicomponent interventions, methodological variability among studies already published and the use of standard outcomes mostly centered on surgical aspects, could be some of the reasons why the evidence regarding the benefits of these programs remains unclear. In the authors’ opinion, CGA should be the first step towards the creation of a multidisciplinary network which would give the patient access to a personalized treatment plan conformed by integral interventions. Although the evaluation of multicomponent programs of these characteristics is difficult, research on this matter seems necessary, as the complexity of elderly patients needs to be confronted not in just one field separately (be it the surgical, clinical, physical, or psychological), but in all of them together.

## Figures and Tables

**Table 1 ijerph-18-06072-t001:** Summary of studies in which the influence of frailty on post-operative outcomes in colorectal cancer populations was assessed.

Author, yr.	Sample (n)	Age (yr.)	Setting	Frailty Measure	Frailty Prevalence	Outcomes & Results
Kristjansson el al. (2010) [69]	178	Mean: 80 SD: 6 Range: 70–94	Hospitalization Elective surgery	CGA (Frail: ≥1 domain affected) − Barthel < 19 − NEADL: NR − CIRS: any grade 4/>2 comorbidity grade 3 − MNA < 17 − MMSE < 24 − Polypharmacy > 7 − GDS > 13	F: 76 (43%) NF: 102 (57%)	30-Day Postoperative Complications Any complication (Clav. I–IV) RR 1.59, 95% CI (1.25–2.01) F: 58 (76%), NF: 59 (48%) Severe Complications (Clav. ≥II) RR 1.75, 95% CI (1.28, 2.41) F: 47 (62%), NF: 36 (35%)
Ommundsen et al. (2014) [70]	178	Age Groups: 70–79 (50%) 80–89 (44%) ≥90 (6%)	Hospitalization Elective surgery	CGA (Frail: ≥1 domain affected) − Barthel < 19 − NEADL: NR − CIRS: any grade 4/>2 co-morbidity grade 3 − MNA < 17 − MMSE < 24 − Polypharmacy > 7 − GDS > 13	F: 76 (43%) NF: 102 (57%)	5-Year Survival F: 18 (24%), NF: 67 (66%) *p* < 0.001
Reisinger et al. (2015) [71]	153 *	>70	Hospitalization Elective surgery	GFI (Frail: ≥5/15) − Mobility − Cognition − Nutrition − Vision − Hearing − Co-morbidity − Physical fitness − Psychosocial	F: 39 (26%) NF: 114 (75%)	Postoperative Sepsis OR 3.96, 95% CI (1.14–13.83) F: 6 (15.4%), NF: 5 (4.4%) *p* = 0.03
Tan et al. (2012) [72]	83	Mean: 81 Range: 75–93	Hospitalization Elective surgery	Fried Criteria (Frail: ≥3/5) − Weigth loss (≥10 lb, >5%) − Physical exhaustion − Physical activity level − Grip strenght − Walking speed	F: 23 (28%) NF: 60 (72%)	Major Postoperative Complications (Clav. ≥ II) OR 4.083, 95% CI (1.433–11.638) F: 11 (47.8%), NF: 11 (18.3%) *p* = 0.006

* Reporting a subgroup of patients that had >70 years, a GFI performed preoperatively and outcomes in the post-operative period. Abbreviations: F: Frail; NF: Not Frail; CGA: Comprehensive Geriatric Assessment; GFI: Groningen Frailty Indicator; Clav.: Clavien-Dindo; OR: Odds Ratio; RR: Relative Risk; CI: Confidence Interval.

**Table 2 ijerph-18-06072-t002:** Summary of studies assessing the benefits of a geriatric liaison on the approach of colorectal cancer patients undergoing surgery.

Author, yr.	Sample (n; Groups)	Age (yr.) ^†^	Design	Type of Program & Setting	Assessments & Interventions	Benefits ^‡^
Shipway et al., 2018 [84]	682 CG: 203 IG: 479	>60 CG: 73 (60–100) IG: 73 (60–94)	Single-center before-after study	Embedded Geriatric Liaison Hospitalization Elective & Emergency GI surgery	Preoperative CGA - Comorbidity - Medication - Nutrition Status - Exercise Tolerance - Cognitive Function - Frailty - Depression Assessment - Functional Capacity - Social Circumstances - Screening Investigations Postoperative Follow-Up - Ward Rounds (selected patients) - Discharge Plan - Geriatrician-led rehabilitation ward (selected patients)	LOS (All Surgeries) All Patients Mean Reduction: 3.1 days 95% CI (0.7–5.5), *p* = 0.007 Patients ≥ 75 years Mean Reduction: 3 days 95% CI (0.2–5.8), *p* = 0.045 LOS (Elective Surgery) All Patients Mean Reduction: 1.3 days 95%CI (−1.4–4.03) Patients ≥ 75 years Mean Reduction: 5.2 days 95% CI (−1.7–12.1), *p* = 0.099
Tarazona- Santabalbina et al., 2019 [86]	310 CG: 107 IG: 203	≥ 70 CG: 75 ± 5 IG: 76 ± 5	Single-center retrospective cohort study	In-Hospital Program Elective CRC Surgery	8-Hour Ward Assessment/intervention - Pressure Ulcers - Pain - Urinary Continence - Constipation - Delirium Daily Ward Assessment - Endovenous Catheters - Medication - Infections - Thromboembolic Events - Anemia - Early Ambulation - Fall Risk - Hydration & Nutrition - Sleep Hygiene - Sensory Impairment	Delirium reduction CG: 31 (29.2), IG: 23 (11.3) *p* < 0.001 Geriatric Syndromes diagnosis * CG: 28 (26.2), IG: 21(10.3) *p* < 0.001 Blood transfusions reduction CG: 24 (22.4); IG: 18 (8.9) *p* = 0.001
Ramirez-Martín et al., 2020 [85]	175 CG: 122 IG: 53	≥80	Single-center before-after cohort study	Emergency Surgery Inpatients - In-Hospital Collaborative Management Elective CRC Surgery - Preoperative CGA & Intervention - In-Hospital Collaborative Management	Preoperative CGA - Not Specified In-Hospital Management - Daily Ward Monitoring - Clinical Management Collaboration - Discharge Planning	LOS (days) Emergency Admissions CG: 27.2 (18.1), IG: 16.6 (10.7) *p* < 0.01 Elective Surgery (CRC) CG: 19.1 (13.4), IG: 10.6 (9.3) *p* < 0.01
Indrakusuma et al., 2015 [87]	100 ** CG: 50 IG: 50	≥70 CG: 75 (71–78) IG: 81 (79–85)	Single-center retrospective cohort and match-control study	Preoperative CGA & Intervention Elective CRC surgery	Preoperative CGA & Intervention - Full Medical Study - Social Circumstances - Review of Systems - Functional Capacity - Family History - Full Physical Examination - Laboratory Tests - Cognitive Function - Depression Assessment - Nutritional Status	Mortality No significant differences Postoperative Delirium No significant differences Postoperative Complications No significant differences Length of Stay No significant differences
Ommundsen et al., 2017 [70]	114 CG: 62 IG: 52	>65 CG: 79 ± 8 IG: 78 ± 7	Multi-center randomized controlled trial	Preoperative CGA & Intervention Elective CRC Surgery - Frail Subjects	Preoperative CGA - Activities of Daily Living - Use of Medication - Comorbidity - Nutritional Status - Cognitive Function - Depression Assessment	Mild Postop. Complications Mentioned. No specific data provided Postop. Complications (Clav. I–V) No significant differences Postop. Complications (Clav. II–V) No significant differences 30-Day Mortality No significant differences 3-Month Mortality No significant differences Length of Stay No significant differences

† Age: mean ± SD; median (interquartile ranges); ‡ Benefits: mean ± SD; absolute number (percentage); ***** ‘Geriatric Syndromes & Events’ include: falls, pain, urinary incontinence, constipation, pressure ulcers, malnutrition and immobility; ****** From this study we only report a subgroup of patients (those in the intervention group that underwent the Geriatric Assessment and their matched controls). Abbreviations: CG: Control Group; IG: Intervention Group; GI: gastrointestinal; CGA: Comprehensive Geriatric Assessment; CRC: Colorectal Cancer; Postop: postoperative; Clav.: Clavien-Dindo.

**Table 3 ijerph-18-06072-t003:** Description of interventions and results of randomized controlled trials in older than 60 years with CRC.

Author, yr.	Sample (n)	Age yr. (mean)	Duration	Features	Programs Analyzed	Principal and Other Outcomes and Results	Adherence
Prehabilitation Programs							
Dronkers et al., 2010 [97]	42	70	2–4 w.	Supervised vs unsupervised home-based exercise program.	Short-term intensive program Group: 2x/w. for 60 min per session. Resistance training, inspiratory muscle training, moderate aerobic training, and training functional activities. Daily 30 min walk and inspiratory muscle training.	Postoperative complications: NSD; LOS: NSD. Functional capacity (TUG): NSD; Strength (CRT): NSD; (Hand grip): NSD. Inspiratory muscle endurance (RMA): SD. Physical activity (LASA): NSD. Fatigue (AFQ): NSD. Quality of life (EORTC-QLQ-C30): NSD.	Supervised training: 97%; Unsupervised training: Non- reported.
					Home-based exercise group: Daily activation for minimally 30 min with a pedometer. Measurements: baseline and postsurgery.		
Carli et al., 2010 [96]	112	60	4 w.	In-hospital supervised exercise program.	Bike/strengthening Group: 7x/w. 20–30 min of moderate intensity aerobic training and 3x/w. for 10–15 min of resistance training.	Functional capacity (6MWT): NSD. Anxiety and Depression (HADS): NSD.	16%
					Walk/breathing Group: 7x/w. training for 40–45 min per session. Walking and breathing exercises.		
					Measurements: baseline, 1-week presurgery, 2- and 4-month postsurgery.		
Minnella et al., 2020 [98]	42	67	4 w.	Prehabilitation Unit-based supervised exercise program.	High-Intensity Interval Training (HIIT) Group: 3x/w. for 40 min per session. High intensity aerobic training on a bicycle and resistance training.	30-day complications (Clav.): NSD. LOS: NSD. Functional capacity (6MWT): NSD; (CPET): SD.	HIIT/MICT: Exercise: 89%/93%, Nutrition: 97%/99%, Loss of patients at follow-up: presurgery 19%/5% 1 month postsurgery 33%/38%, and 2 months postsurgery. 38%/48%.
					Nutrition: Balanced macronutrient composition and protein.		
					Anxiety: Relaxation techniques and breathing exercises.		
					Moderate Intensity Continuous Training (MICT) group: 3x/w. for 50 min per session. Moderate intensity aerobic training on a bicycle and resistance training.		
					Nutrition: Balanced macronutrient composition and protein Anxiety: Relaxation techniques and breathing exercises. Measurements: baseline, presurgery, 1- and 2-month postsurgery		
Berkel et al., 2021 [95]	57	74	3 w.	Community-based supervised exercise program.	Intervention group: 3x/w. for 60 min per session.	30-day postoperative complications (Clav.): SD. LOS: NSD. Readmissions: NSD. Functional capacity- Aerobic fitness (V02): NSD.	Non- reported.
					Moderate to high intensity aerobic training on a cycle ergometer and resistance training.		
					Control group (usual care): ERAS protocol. Nutritional counseling and advice on smoking cessation. Measurements: 30-day postsurgery.		
Chia et al., 2015 [100]	117	80	PREHAB: 2w. and REHAB: 2–6 w.	Supervised exercise or unsupervised home-based exercise depending of patient situation.	Intervention group (STF): Cardiovascular strengthening, mobilizing and muscle strengthening. Nutrition: Individual attention.	Postoperative complications (Clav.): NSD. LOS: SD. 30-day mortality: NSD. Recovery functional capacity (Barthel): NSD.	80%
					Control group (GSS): quality reviews and a patient-centered culture. Nutrition: Individual attention.		
					Measurements: 30-day mortality and 6-week postsurgery.		
Awasthi et al., 2018 [99]	140	68	PREHAB: 4 w. and REHAB: 8 w.	Supervised exercise and unsupervised home-based exercise program.	Group 1: unsupervised exercise (Gillis et al. 2014) 3x/w. for 50 min per session: moderate intensity aerobic exercise and moderate intensity resistance training. Nutrition: Nutritional assessment and protein supplementation. Anxiety: Relaxation techniques.	LOS: NSD Functional capacity (6MWT): SD. Muscle strength (Hand grip): SD. Anxiety and depression (HADS): NSD. Quality of life (SF36): SD.	Supervised exercise: 98%
					Group 2: supervised training (Bousquet-Dion et al., 2018) 1x/w. at hospital exercise laboratory for 65 min per session: moderate aerobic exercise and resistance exercise. 3 to 4x/w. at home for 30 min of moderate intensity aerobic training and resistance training.		
					Nutrition: Nutritional assessment and protein supplementation.		
					Anxiety: Relaxation techniques.		
					Measurements: baseline, before surgery, 4- and 8-week postsurgery.		
Souwer et al. 2018 [101]	86	81	PREHAB: 4 w. and REHAB: 4–6 w.	In-hospital supervised exercise and home-based exercise program.	Intervention Group: 2x/w. for 30–45 min per session. PREHAB: Resistance and endurance training, and home exercise and breathing. REHAB; Physical training. Nutrition: Protein supplementation, dietary support. Anxiety: Cognitive and emotional guidance.	30-day postoperative complications. Only cardiac: SD. LOS: SD. 1-year mortality: NSD.	63%
					Control group: usual care (previous cohort).		
					Measurements: 30-day and 1-year postsurgery.		
Gillis et al., 2014 [104]	77	66	PREHAB: 4 w. vs REHAB: 8 w.	Unsupervised home-based exercise program.	PREHAB: 3x/w. for 50 min per session. Moderate intensity aerobic exercise and moderate resistance training. Nutrition: Protein intake. Anxiety: 60 min of relaxation techniques and breathing exercises.	30-day complications (Clav.): NSD. Functional capacity (6MWT): SD. Health status (SF-36): NSD. Anxiety and depression (HADS): NSD.	PREHAB = 78%; 4 w.: 53%; 8w.: 53% REHAB = 4 w.: 31% 8w.: 40%
					REHAB Group: 3x/w. for 50 min per session. Moderate intensity aerobic exercise and moderate resistance training. Nutrition: Protein intake. Anxiety: 60 min of relaxation techniques and breathing exercises.		
					Measurements: baseline, presurgery, 4- and 8-week postsurgery.		
Minnella et al., 2017 [105]	185	68	PREHAB: 4 w. vs REHAB: 8 w.	Unsupervised home-based exercise and supervised exercise program.	PREHAB Group: 3x/w. for 20–30 min per session of endurance training and 2x/w. of resistance training. Nutrition: Dietary changes and protein supplementation. Anxiety: Relaxation techniques. Plus ERAS protocol.	Postoperative complications (Clav.): NDS. LOS: NSD. Functional capacity (6MWT): SD. Physical fitness (CHAMPS): SD.	PREHAB = 70–98% REHAB = 4w.:53–72% 8 w.: 53–82%
					REHAB Group: 3x/w. for 20–30 min per session of endurance training and 2x/w. of resistance training. Nutrition: Dietary changes and protein supplementation. Anxiety: Relaxation techniques. Plus ERAS protocol.		
					Measurements: baseline, presurgery, 4- and 8-week postsurgery.		
Bousquet-Dion et al., 2018 [102]	80	73	PREHAB: 4 w. vs REHAB: 8 w.	In-hospital supervised and unsupervised home-based exercise program.	PREHAB Group: At hospital: 1x/w. for 65 min per session. Moderate aerobic exercise and resistance exercise. At home: 3 to 4x/w. for 30 min of moderate intensity aerobic training and resistance training. Nutrition: Protein supplementation. Anxiety: 60 min of relaxation techniques and breathing exercises. Plus ERAS protocol.	LOS: NSD. Functional capacity (6MWT): NSD. Physical activity (CHAMPS): NSD.	Supervised exercise; PREHAB 98% REHAB = 4 w. 70% 8 w.: 75%
					REHAB Group: At hospital: 1x/w. for 65 min per session. Moderate aerobic exercise and resistance exercise. At home: 3 to 4x/w. for 30 min of moderate intensity aerobic training and resistance training. Nutrition: Protein supplementation. Anxiety: 60 min of relaxation techniques and breathing exercises. Plus ERAS protocol. Measurements: baseline, presurgery, 4- and 8-week postsurgery.		
Carli et al., 2020 [103]	110	78	PREHAB: 4 w. vs REHAB: 4 w	Unsupervised home-based multimodal program with 1 session /w. supervised at hospital (similar for PREHAB and REHAB).	PREHAB group: 1x/w. for 60 min per session. Supervised moderate aerobic exercises and resistance exercises at hospital and moderate intensity aerobic activities (walking) and resistance training at home. Nutrition: Protein intake. Anxiety: Relaxation techniques and breathing exercises.	30-day postoperative complications (CCI and Clav.): NSD. LOS: NSD. Functional capacity (6MWT): NSD. Readmissions: NSD.	PREHAB = 68% REHAB = 14%. General exercise and nutrition: PREHAB 80%, REHAB 30%.
					REHAB group: 3x/w. for 30 min per session. Supervised moderate aerobic exercises and resistance exercises at hospital and moderate-intensity aerobic activities (walking) and resistance training at home. Nutrition: Protein intake. Anxiety: Relaxation techniques and breathing exercises. Measurements: 4-week postsurgery.		

Abbreviations: NSD = Non Significant Differences between groups; SD = Significant Differences between groups; LOS = Length of hospital Stay; TUG = Timed Up and Go; CRT = Chair Rise Time; RMA = Respiratory Muscle Analyzer; LASA = Physical Activity Questionnaire; AFQ = Abbreviated Fatigue Questionnaire; EORTC-QLQ C30 = The EORTC Quality of Life questionnaire QLQ-C30; 6MWT = 6-Minute Walking Test; HADS = Hospital Anxiety and Depression Scale; Clav. = Clavien-Dindo; CPET = Cardiopulmonary Exercise testing; VO2 = máx O_2_ Volume; PREHAB = Prehabilitation; REHAB = Rehabilitation; SF36 = Short Form (36); STF = Trans-institutional transdisciplinary Start to Finish Programm; GGS = Geriatric Surgery Service; Health Survey; CHAMPS = Community Health Activities Model Program for Seniors; ERAS = Enhanced Recovery After Surgery; CCI = Comprehensive Complications Index; yr. = years; w. = weeks; min = minutes.

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
