# Peer review of "Colorectal Cancer in Elderly Patients with Surgical Indication: State of the Art, Current Management, Role of Frailty and Benefits of a Geriatric Liaison"

_ijerph, 2021, doi:10.3390/ijerph18116072_

Round 1

Reviewer 1 Report

ijerph-1223031-v1

Title: Colorectal Cancer in Elderly Patients with Surgical Indication: State of the Art, Current Management, Role of Frailty and Benefits of a Geriatric Liaison

Corresponding author: Luis M. Jiménez-Gómez María T. Vidan-Astiz

Even though authors reviewed it properly, there are several places to make further corrections. 

Specific comments are as follows:

  1. This paper has reviewed the CRC of the elderly from a wide variety of angles, and there is insufficient review on the surgical results according to the histological aspect of CRC. For CRC, the most common form of colon cancer is adenocarcinoma, constituting between 95% to 98% of all cases of CRC. Other, rarer types include lymphoma, adenosquamous and squamous cell carcinoma. Therefore, as far as possible, we ask you to review the surgical results according to histological aspects.
  2. Lines 52, 134, 453, and more: When writing 'e.g.', always use the same pattern.
  3. Lines 213 and 272: ‘2349’ should be ‘2,349’. ‘1967’ should be ‘1,967’.
  4. Lines 238, 387, 391, and Title of Table 3: CCR should be CRC.
  5. Table 1: Frailty prevalence should be Frailty Prevalence.
  6. Table 1: 10lb should be 10 lb.
  7. Table 1: p0.006 should be p = 0.006.
  8. Line 364: Please define ICU.
  9. Line 411: RTC should be RCT.
  10. Table 2: Urinaty Continence should be Urinary Continence.
  11. Table 2: Tromboembolic Events should be Thromboembolic Events.
  12. Table 3: NDS should be NSD.

Overall, the manuscript can be considered to publication after minor revision as indicated above.

Reviewer 2 Report

This manuscript provides a useful review of the management of older patients with colorectal cancer. While the review is interesting and comprehensive, some grammatical errors and typos are affecting the flow of the work. The manuscript needs English editing. Moreover, the authors should address the following points:

Abstract:

There is nothing on frailty in the abstract even though it was in the topic.

Introduction:

P 3 L106: I do not think it is appropriate to refer to (<75) as young. Reword, please.

L 122: “ colon LS patients had a shorter time until fluid and solid intake initiation” I do not quite understand what the point is. Reword, please.

L 191: provide a definition of functional dependency.

L 212: provide a reference for frailty prevalence.

L212 -213: “Cancer increases the 212 prevalence of frailty: 46% to 80% in a sample of 2349 Medicare beneficiaries”

The sentence is not clear. elaborate on this, please.

L214: “ The association of both cancer and frailty increases the risk of poor evolution and treatments 214 adverse events.” Do you mean the association between cancer and frailty?

L 230: “Robinson et al. in a prospective cohort study that included patients 65 years undergoing elective colorectal or cardiac surgeries, described how those who met at least 4 of 7 frailty-related characteristics (regarding function, cognition, chronic disease burden, walking speed, nutrition and geriatric syndromes) had a higher risk of developing post-operative complications. “

Specify which frailty definition these criteria were related to.

Table 1:

Add the GFI, OR, RR in the acronyms.

It would also be useful to give an overview of the main findings of these studies in the text.

L 350: elaborate on the observed trend.

Round 2

Reviewer 2 Report

I would like to thank the authors for incorporating my comments within the revised version of the manuscript. The manuscript has significantly improved. I have no further comment to add.